# Full-Body Harness versus Waist Belt: An Examination of Force Production and Pain during an Isoinertial Device Maximal Voluntary Isometric Contraction

**DOI:** 10.3390/jfmk9030165

**Published:** 2024-09-16

**Authors:** Trevor J. Dufner, Jonathan M. Rodriguez, McKenna J. Kitterman, Jennifer C. Dawlabani, Jessica M. Moon, Adam J. Wells

**Affiliations:** Exercise Physiology Intervention and Collaboration Lab, School of Kinesiology and Rehabilitation Sciences, University of Central Florida, Orlando, FL 32816, USA; trevor.dufner@ucf.edu (T.J.D.);

**Keywords:** MVIC, pain, force, squat, isometric, isoinertial

## Abstract

Background/Objectives: This study examined the differences in participant force production and pain between a squat maximal voluntary isometric contraction (IMVIC) performed with either a waist belt (WB) or full-body harness (FBH) on the Desmotec D.EVO isoinertial device (D.EVO). Agreement between FBH IMVIC and a traditional force plate squat MVIC (TMVIC) was also assessed. Methods: Twenty adults completed FBH, WB, and TMVIC assessments on two separate occasions. Two-way treatment x time ANOVAs were conducted to compare force outputs and pain between treatments (FBH vs. WB) across time. Test-retest reliability was assessed using intraclass correlation coefficients. Associations between outcomes were determined using Pearson’s r. Standard error of estimate, constant error, total error, and Bland-Altman plots were used to assess agreement between IMVIC and TMVIC. Results: FBH and WB IMVIC exhibited good to excellent reliability (ICC_2,1_ = 0.889–0.994) and strong associations (r = 0.813 and 0.821, respectively) when compared to TMVIC. However, agreement between FBH and TMVIC was poor. No significant interaction or main effects were observed for pain. FBH maximum isometric force (MIF) was significantly higher than WB MIF. WB IMVIC was the only significant predictor of TMVIC (R^2^ = 0.674). Conclusions: Our findings indicate that the D.EVO should not be utilized as a replacement for a traditional MVIC setup.

## 1. Introduction

Muscular strength is defined as the ability of a muscle or muscle group to produce force against resistance [1]. In this regard, maximal voluntary isometric contractions (MVICs) are often utilized to assess force production as they are relatively low-risk, simple to perform, and typically exhibit high test-retest reliability compared to dynamic assessments [1]. MVICs assessed via isokinetic and force plate technologies [1,2,3] are considered to be the most valid non-invasive assessment of muscular damage [4] and are commonly used to assess progressions of training [1], neuromuscular disease [5,6], and post-op rehabilitation [7,8]. More recently, isoinertial devices have also been utilized to assess MVIC. Typically, these devices are utilized during dynamic movements for the purposes of eliciting greater eccentric loads relative to traditional isotonic movements [9,10,11]. Users are fitted with either a full-body harness (FBH: Figure 1) or a waist belt (WB: Figure 2) connected to a weighted flywheel via a pulley mechanism, which returns the force generated during the concentric phase of a movement as an equal and opposite force during the eccentric portion of the movement [9,10]. During an MVIC, however, the flywheel is locked, allowing the user to produce maximal upward force in a manner similar to a traditional high-bar back squat. Notwithstanding, anecdotal reports of discomfort and pain have been noted in our lab due to compression of the anterior portion of the hips by the WB during completion of the IMVIC. Furthermore, Layer et al. [12] previously examined the kinetic differences between the belt squat and barbell variants of a squat MVIC and observed greater force production during the belt squat. As such, it would be more appropriate to only compare squat MVIC variants with variants that are analogous in both kinematic and biomechanic requirements. Therefore, considering the FBH and WB straps are designed to elicit different loading on the participants, which may consequentially elicit different kinematic and biomechanic requirements, an investigation examining the possible differences in force production and subsequent relationship with a more traditional squat MVIC (TMVIC) is warranted.

Physiological pain can be defined as an unpleasant sensory and emotional experience associated with actual or potential tissue damage [13] and is generally considered to serve two biologically useful purposes within the body: one of reactance to a noxious stimulus, and the other of deterrence from a potential future noxious stimulus or intensification of one already occurring [14]. According to the biopsychosocial model, pain is an experience influenced by biological, psychosocial, and social/environmental factors rather than being exclusively biological in nature [15]. Thus, even when presented with matched stimuli, individual responses (i.e., modulation of movement) may vary between individuals [16]. A number of previous investigations have reported muscular dysfunction and adaptations in motor strategies in response to induced pain [14,17,18]. Hug et al. [14] assessed force production during a bilateral squat MVIC before and after experimentally inducing pain with a hypotonic saline solution in the left vastus medialis of 15 young adults. They observed significantly reduced force productions in the left (painful) leg of ~32.6% and ~10.0%, when compared to the right (non-painful) leg and to the control condition, respectively. Park and Hopkins [18] examined the effects of experimentally inducing anterior knee pain on involuntary and voluntary quadriceps activation during a knee extension MVIC in 13 healthy college students. They observed involuntary and voluntary reductions in quadriceps activation of 12% and 34%, respectively, immediately after injection of a saline solution into the lateral infrapatellar fat pad of the participants’ dominant limb. These findings led Park and Hopkins [18] to conclude that a combination of both inhibitory mechanisms, such as a spinal reflex, associated with higher brain center voluntary mechanisms and involuntary inhibitory mechanisms were responsible for the reduction in quadriceps activation [18]. Together, these investigations [14,18] indicate that when presented with noxious stimuli during muscular contraction, individuals employ voluntary and involuntary strategies to reduce force production as a means of mitigating the intensification of the painful stimulus. Consequentially, force values achieved during maximal muscle contraction assessments (i.e., WB IMVIC) may not be exclusively representative of absolute strength but instead illustrate a composite measure of strength, the pain induced by the assessment itself, and the pain tolerance of the individual performing it. Thus, pain may need to be accounted for when assessing maximal muscular strength during IMVICs.

Several studies to date have investigated the effects of pain, body position, knee angle, and bar position on muscle activation and force production during a squat [1,14,19,20,21,22,23,24]. However, to our knowledge, no studies exist examining the differences in force output during FBH and WB IMVIC’s on the D.EVO or the reliability and concurrent validity of a squat FBH IMVIC itself. Considering a belt squat variant has been previously shown to elicit greater force production than a barbell variant, the reports of pain experienced during WB IMVICs within our own laboratory, and pain’s modulatory role in force production, an investigation examining the differences in pain, force output, test-retest reliability, and concurrent validity of an FBH IMVIC when compared to a WB IMVIC and TMVIC is warranted. Therefore, the purpose of this investigation was to examine differences in participant pain and force production during FBH and WB MVICs on the D.EVO. Additionally, we sought to examine the reliability and concurrent validity of an FBH IMVIC and further examine these indices for the WB IMVIC when compared to TMVIC.

## 2. Materials and Methods

### 2.1. Experimental Design

Participants reported to the Exercise Physiology Intervention and Collaboration (EPIC) lab for a total of 5 visits, including 1 screening visit (visit 1; V1), 2 familiarization visits (visits 2 and 3; [V2 and V3]), and 2 testing sessions (visit 4 and visit 5; [V4 and V5]). This study utilized a within-group repeated measures design with randomization of strap order to evaluate the effects of different IMVIC protocols (FBH and WB) repeated during two different testing sessions (V4 and V5).

#### 2.1.1. Screening

During the screening visit (V1), subjects provided written informed consent and completed a physical activity readiness questionnaire (PAR-Q+), medical history questionnaire (MHQ), caffeine consumption questionnaire (CCQ), and if applicable, a menstrual status questionnaire (MSQ) to determine eligibility. Eligible participants were then randomized in a counterbalanced fashion into either FBH IMVIC first (WB IMVIC second) or WB IMVIC first (FBH IMVIC second) groups using an online randomization tool. Traditional MVIC (TMVIC) was completed following the FBH and WB IMVICs, regardless of group assignment.

#### 2.1.2. Familiarization

At least 24 h after V1, participants returned for the first of two familiarization visits, V2, where they completed anthropometric assessments (height, weight, and body composition via bioelectrical impedance analysis [BIA]). Participants then underwent a brief standardized warm-up and were familiarized with the FBH and WB IMVIC protocols in an order designated by their randomized group assignment before being familiarized with the TMVIC. At least 24 h later, participants returned to the lab for V3, where they completed a warm-up identical to that performed during V2, followed by additional familiarization with FBH, WB, and TMVIC assessments.

#### 2.1.3. Testing

Twenty-four to 72 h following V3, participants returned to the lab to complete the first of 2 testing sessions (Visit 4 & Visit 5). The testing sessions occurred in an identical fashion to one another, which consisted of the same standardized warm-up performed during V2, completion of the FBH and WB IMVIC protocols in their assigned order, and completion of the TMVIC protocol, respectively. V2–V5 were scheduled for the same time of day as V1 and were separated by 24–72 h. MVIC assessments during V2–V5 were separated by a 5 min rest to allow for adequate recovery. Participants were required to abstain from any strenuous lower-body exercise that could result in soreness for the entirety of the study.

### 2.2. Participants

A power analysis using power analysis software (G*Power 3.1.9.4, HHU, Dusseldorf, Germany) revealed that to detect a moderate Pearson’s correlation (r = 0.60) with a power of 0.80 and *p*-value of 0.05, a sample size of 19 would be required. Accordingly, a total of 20 (22 ± 3 yrs, 165.99 ± 7.45 cm, 73.50 ± 17.27 kg, 22.09 ± 9.81 BF%) recreationally active men (8) and women (12) completed the study protocol. Prior to completing any study-related procedures, potential participants were informed of the potential benefits and risks of the study, provided written informed consent, and completed PAR-Q+, MHQ, and CCQ questionnaires to determine eligibility. Eligibility criteria included at least 150 min of combined physical activity per week according to the ACSM standard for recreationally active individuals. Participants were required to be healthy and ready for activity as determined by a PAR-Q+ and MHQ, report no current or prior use of any performance-enhancing drugs, and be free from any previous or current lower body injuries that were viewed by the investigators to potentially limit the ability of the participant to perform the required assessments. Female participants were required to be premenopausal as determined by the MSQ. Individuals reporting habitual daily consumption of >300 mg of caffeine as determined by a CCQ were not permitted to continue with the study.

### 2.3. Procedures

#### 2.3.1. Anthropometrics

Height, weight, and body composition were assessed during V2. Height and weight were assessed using a stadiometer and scale (Health-o-meter Professional Patient Weighing Scale, Model 500 KL, Pelstar, Alsip, IL, USA), and body composition was assessed via BIA (InBody 770, Biospace Co., Ltd., Seoul, Republic of Korea). For the BIA assessment, participants were asked to be at least two hours fasted and well hydrated. Prior to testing, participants were asked to void their bladder and remove shoes, socks, and all jewelry. Next, participants were instructed to stand barefoot on the BIA device, to grab hold of the two handheld electrodes, and to remain still and quiet for approximately 30 s while their body composition was calculated.

#### 2.3.2. Maximum Voluntary Isometric Contractions (MVIC)

FBH and WB MVIC assessments were completed using the D.EVO (D.EVO, Desmotec, Biella, Italy). MVIC force (Newtons: N) was measured via two-underfoot load cells at an effective sampling rate of 25 Hz. Participants were secured onto the isoinertial device in a squat position via a flywheel cable attached to a waist belt in accordance with the manufacturer’s recommendations. Cable length for all IMVIC’s was established during V2, utilizing a maximal isometric contraction that elicited a knee angle of 140° (measured via goniometer). The maximal contraction ensured that the knee angle was consistent between WB and FBH MVIC and TMVIC protocols by accounting for the stretch of the fabric waist belt and compression of the surrounding tissue. For each FBH and WB MVIC, participants were instructed to place their feet at a width identical to that which they would typically use during a traditional back squat, to cross their arms in front of them across their chest, and to replicate that position for each IMVIC.

TMVIC assessments were completed using dual force plate sampling at 600 Hz (Accupower, AMTI, Watertown, MA, USA) inside a customized squat rack and platform with an immovable bar. Bar height for all TMVICs was established for each participant during V2, utilizing a high-bar position that elicited a knee angle of 140° (measured via goniometer) and was standardized for all subsequent TMVICs. For each TMVIC, participants were instructed to drive upwards against the bar by attempting to extend their knees and hips maximally in a manner identical to the movement pattern of a typical high bar back squat.

Each MVIC assessment consisted of 3 × 5 s maximal isometric contractions. Each contraction was initiated following a verbal prompt from the researcher, and consistent verbal encouragement was provided for the entire 5 s of each contraction. One minute of rest was provided between each contraction to allow for recovery. The contraction containing the highest peak force during each assessment was denoted as the participant’s maximum isometric force (MIF), and the average of the peak force achieved during each of the 3 contractions within an assessment was denoted as the average peak isometric force (APIF). FBH, WB, and TMVIC were assessed during testing sessions 1 and 2 (V4–V5).

#### 2.3.3. Pain

The participant’s perceived pain was assessed using a 100 mm visual analog scale (VAS). The scale consisted of a 100 mm line, with 0 indicating “no pain at all” and 100 indicating “maximal or excruciating pain”. Participants were asked to rank the level of pain imposed by the FBH or WB during each contraction immediately following its completion by drawing a mark on the VAS with a pen that coincided with the magnitude of pain felt. The furthest distance from the 0 in mm was used to denote the participant’s maximum pain (MP), and the average distance in mm for all three contractions was denoted as the participant’s average pain (AP). Subjective pain was assessed during testing sessions 1 and 2 (V4–V5).

### 2.4. Statistical Analysis

Prior to analyses, data were assessed for normality using the Shapiro-Wilks test. Two-way treatment x time repeated measures ANOVAs were conducted to compare force outputs and pain between treatments (FBH vs. WB) across time (V4 vs. V5). If the assumption of sphericity was violated, a Greenhouse-Geiser correction was applied. Where a significant interaction occurred, simple main effects were utilized to assess the interaction where appropriate. Where there was no significant interaction between treatments, main effects were reported. Effects were further analyzed using partial eta-squared (η_p_^2^) and Hedges’ *g* (*g*) effect sizes. Partial eta-squared was evaluated in accordance with Cohen [25] at the following levels: small (0.01–0.058), medium (0.059–0.137), and large (>0.138) effects. Hedges *g* was interpreted using thresholds of <0.2, 0.2 to <0.6, 0.6 to <1.2, 1.2 to <2.0, and 2.0 to 4.0, which correspond to trivial, small, moderate, large, and very large ES, respectively. The test-retest reliability between sessions was assessed using intraclass correlation coefficients (ICC). ICC estimates and their 95% confident intervals were calculated using SPSS (statistical package version 28.0.1.0, SPSS Inc., Chicago, IL, USA) based on an absolute-agreement, 2-way mixed-effects model and interpreted as follows: >0.90 = excellent; <0.90 to 0.75 = good; <0.75 to 0.50 = moderate; <0.50 = poor, and interpreted based on their 95% confidence interval (CI) [26]. Standard error of the measurement (SEM) and minimum difference (MD) values were also calculated. Pearson’s correlation coefficients (r) were used to assess the associations between TMVIC, FBH IMVIC, and WB IMVIC for MIF and APIF. Pearson correlation coefficients were interpreted as <0.1 = trivial; 0.1 to <0.3 = small; 0.3 to <0.5 = moderate; 0.5 to <0.7 = large; 0.7 to <0.9 = very large; and 0.9 to 1.0 = almost perfect [25,27]. All significant correlations were subsequently entered into a stepwise regression to determine which significantly associated outcomes best predicted TMVIC performance. Paired sample *t*-tests were used to determine the difference between TMVIC APIF and FBH and WB APIF during visit 5 (testing session 2). Validation of the IMVIC (FBH and WB) during visit 5 was based on the evaluation of the criterion (TMVIC) vs. FBH and vs. WB via calculation of the constant error (CE), Pearson’s product-moment correlation (r), standard error of estimate (SEE), and total error (TE) [28]. The Bland-Altman approach [29] was also used to assess agreement between the methods. (TMVIC vs. FBH and TMVIC vs. WB). All statistical procedures were accepted as significant at an alpha level of *p* ≤ 0.05. All statistical analyses were completed using SPSS statistical software (v. 28.0.1.0, SPSS Inc., Chicago, IL, USA).

## 3. Results

### 3.1. Isometric Force

Performance data for isometric force production from the two testing sessions (V4 and V5) are reported in Table 1.

No significant treatment x time interaction (F_1,19_ = 0.201, *p* = 0.659, η_p_^2^ = 0.010) or main effect for time (F_2,38_ = 0.171, *p* = 0.684, η_p_^2^ = 0.009) were observed for FBH and WB MIF. A main effect for treatment was observed (F_1,19_ = 4.429, *p* = 0.049, η_p_^2^ = 0.189). Collapsed across time, FBH MIF was significantly higher than WB MIF (*p =* 0.049, *g* = 0.228).

No significant treatment x time interaction (F_1,19_ = 0.011, *p* = 0.919, η_p_^2^ = 0.001), main effect for treatment (F_1,19_ = 4.087, *p* = 0.58, η_p_^2^ = 0.177), or main effect for time (F_1,19_ = 0.235, *p* = 0.633, η_p_^2^ = 0.012) were observed for FBH and WB APIF.

### 3.2. Pain

Participant’s pain data from the two testing sessions (V4 and V5) are reported in Table 1.

No significant treatment x time interaction (F_1,19_ = 0.185, *p* = 0.672, η_p_^2^ = 0.010), main effect for treatment (F_1,19_ = 1.451, *p* = 0.243, η_p_^2^ = 0.071), or main effect for time (F_1,19_ = 0.102, *p* = 0.753, η_p_^2^ = 0.005) were observed for MP. Absolute differences of 5.25 and 3.65 were observed for MP between treatments during V4 and V5, respectively.

No significant treatment x time interaction (F_1,19_ = 0.072, *p* = 0.791, η_p_^2^ = 0.004), main effect for treatment (F_1,19_ = 1.892, *p* = 0.185, η_p_^2^ = 0.091), or main effect for time (F_1,19_ = 0.231, *p* = 0.637, η_p_^2^ = 0.012) were observed for AP. Absolute differences of 4.90 and 3.98 were observed for AP between treatments during V4 and V5, respectively.

### 3.3. Reliability Analysis

Reliability data for the two testing sessions (V4 and V5) are shown in Table 2. FBH APIF and WB MIF and APIF variables exhibited excellent reliability (ICC’s = 0.920–0.994) between testing sessions 1 and 2, while FBH MIF variables exhibited good to excellent reliability (ICC = 0.889–0.982).

### 3.4. Associations between FBH and WB MVIC with TMVIC Performance

Associations between TMVIC, FBH IMVIC, and WB IMVIC for MIF and APIF during testing session 2 (V5) are reported in Table 3. Pearson’s correlation analysis revealed a very large (r = 0.813–0.830) strength of associations for MIF and APIF between TMVIC, FBH IMVIC, and WB IMVIC. Stepwise regression results are shown in Figure 3. WB MIF was the only significant predictor of TMVIC MIF (R^2^ = 0.674, SEE = 46.694), and WB APIF was the only significant predictor of TMVIC APIF (R^2^ = 0.688, SEE = 42.938). No other outcomes significantly predicted TMVIC MIF of APIF.

### 3.5. Validity

Cross-validation analyses of APIF values are shown in Table 4. The coefficients of validity for FBH and WB were relatively strong (r = 0.819 and 0.830, respectively), and the SEE% were both within the acceptable limit (10.0%) [28,30]. It has been previously suggested that TE% is the best indicator of the accuracy of prediction and that valid predictions will exhibit similar SEE and TE values [28]. However, within the current investigation, we observed TE% for the FBH and WB that were outside the acceptable limit (26.03% and 22.44%, respectively) and observed relatively large differences between our SEE and TE values. Furthermore, FBH and WB APIFs were significantly higher than TMVIC APIFs (−695 ± 540 and −509 ± 508, respectively), indicating a consistent bias for both the FBH and WB to overestimate an individual’s squat MVIC force.

The Bland-Atman plots for FBH (Figure 4A) and WB (Figure 4B) APIF are displayed in Figure 4. The regression line (grey) for both FBH and WB (R^2^ = 0.158, *p* = 0.08 and R^2^ = 0.129, *p* = 0.121, respectively) showed no significant proportional bias. However, 95% limits of agreement ranged from −1753 N to 363 N for FBH and from −1504N to 486 N for WB, indicating a high degree of inconsistency and variability between the D.EVO IMVICs and TMVIC. These findings suggest that regardless of strap selection, an IMVIC assessment performed on the Desmotec D.EVO is not predictive of isometric squat performance using the traditional force plate and barbell setup (TMVIC).

## 4. Discussion

The results of this study indicate that both the FBH and WB are reliable straps for assessing squat IMVIC at a knee angle of 140° on the D.EVO and exhibit strong associations with a TMVIC with a matched knee angle. While MIF values were significantly higher for FBH when compared to the WB, this did not appear to be related to perceived pain, which was not significantly different between FBH and WB. MIF and APIF variables for both straps exhibited a very large strength of associations with TMVIC variables (r = 0.813–0.830). Nevertheless, only WB MIF and APIF variables significantly predicted TMVIC MIF and APIF. Furthermore, neither FBH nor WB IMVIC appear to be a valid predictor of TMVIC force. Collectively, these findings indicate that the D.EVO is efficacious for reliably assessing peak IMVIC force; however, strap selection may influence absolute isometric force values. Additionally, the effective sampling rate of the D.EVO remains relatively low (25 Hz), which limits its utility for assessing additional force variables like rate of force development. Accordingly, other force monitoring technologies may be best suited for this purpose.

The current investigation is the first to our knowledge to examine the difference between FBH and WB IMVIC force and strap-imposed pain on the D.EVO. Previous investigations by Hug et al. [14] and Park and Hopkins [18] have shown that when presented with a noxious stimulus during a muscular contraction, participants engage in both voluntary and involuntary motor-modulatory strategies intended to reduce pain intensification. Specifically, Hug et al. [14] observed compensatory motor strategies during a bilateral isometric squat exercise whereby participants disproportionally loaded their unaffected (non-painful) limb to reduce pain in the affected (painful) limb. In contrast, pain associated with the FBH or WB would have likely been distributed across the entire anterior portion of the participants’ hips in a manner proportional to the total absolute force produced during the IMVIC. As such, disproportionate loading would not have been an effective pain-alleviating strategy; rather, the presence of pain may have manifested as an absolute decrease in force output. Consistent with this, MIF values were significantly greater for FBH compared to WB, with no significant differences in perceived pain between the two. This may indicate that any pressure or pain elicited by the FBH may have been more tolerable at any given absolute force output and explain why greater MIF values were achieved. Accordingly, IMVIC MIF values may not be exclusively representative of participants’ maximal isometric squat strength and instead may represent a hybrid of pain tolerance and force-generating capacity. Axial loading has been previously shown to affect force production during isometric squat variants. Layer et al. [12] observed significantly higher peak force production during an isometric belt squat when compared to an isometric barbell squat. Accordingly, the axial loading elicited by the FBH harness may be expected to diminish force production when compared to the WB. Alternatively, the axial loading of the FBH likely elicited kinematic and biomechanic requirements more analogous to the TMVIC. Furthermore, it has been previously shown that the low-bar back squat position evokes different joint angles than that of a high-bar back squat position, which may subsequently increase its load-bearing capacity [21]. Within the current investigation, the over-the-shoulder straps of the FBH elicited a forward leaning posture during the IMVIC as opposed to the upright posture of the WB IMVIC. Although slight, this deviation in joint angles may have made the FBH IMVIC more closely representative of a low-bar back squat and explain why the FBH MIF was greater than the WB MIF. Together, these items may have limited the ability of the FBH IMVIC to predict force in the TMVIC assessment, which was more akin to a high-bar squat. However, considering these joint angle deviations were very slight and that the loading of the FBH was dispersed between the hips and the shoulders and not entirely axial, we hypothesize that the increased participant tolerance of the pain elicited by the FBH was primarily responsible for the elevated force outputs observed during a FBH IMVIC.

This study is not without its limitations. No alternative measure of participant pain tolerance was included within the current investigation. Consequentially, we are unable to fully characterize whether participants’ IMVIC force production was truly representative of their force-generating capacity and not confounded by pain tolerance. Additionally, participants may have employed small deviations in body position between repetitions as a strategy to alleviate strap-imposed pain. Therefore, considering force production during a squat has been shown to be affected by changes in body position and joint kinematics, [1,14,19,20,21,22,23,24] the absence of an objective measure for body position may have affected force outcomes. Furthermore, we did not assess the relative distribution of load between the shoulders and hips during an FBH. As such, we cannot fully characterize the degree to which axial loading affected our FBH outcomes. TMVIC always occurred after both FBH and WB IMVIC. Consequentially, small post-activation potentiation (PAP) or fatigue effects are possible within the current examination. However, we do not believe they are of concern within the present study aim. FBH and WB orders were randomized and counterbalanced. Therefore, even if PAP or fatigue effects were present and augmented TMVIC force, this would not affect the TMVIC’s comparison to the FBH and WS differently.

## 5. Conclusions

In conclusion, both FBH and WB exhibited good to excellent test-retest reliability when assessed within 24–72 h. Additionally, no significant differences in perceived pain were observed between FBH and WB when assessing IMVIC on the D.EVO. However, FBH and WB were not valid predictors of TMVIC performance. Furthermore, the findings of the current investigation further exhibit that comparisons between squat MVIC variants should be interpreted with caution if kinematic and biomechanic requirements are not closely matched. It is likely that the FBH elicited less pain than the WB at any given absolute force; nevertheless, WB IMVIC variables were the only significant predictors of TMVIC variables. Collectively, our findings indicate that the D.EVO device is capable of assessing peak isometric force during a squat at a knee angle of 140° reliably between testing sessions. However, considering FBH and WB outcomes were not valid predictors of TMVIC, future outcomes recorded on the D.EVO should only be utilized to assess changes over time rather than being interpreted as a measure of participants true squat MVIC force. Further, in consideration of the pain imposed by the straps during an IMVIC assessment and the device’s relatively low sampling rate (25 Hz), we do not recommend this device as a replacement for more traditional force plate technologies at this time. Moreover, we recommend that prior to acquiring an analogous device for the purposes of research or performance tracking, potential buyers should acquire assurances that the limitations of the D.EVO are not ubiquitous.

## Figures and Tables

**Figure 1 jfmk-09-00165-f001:**
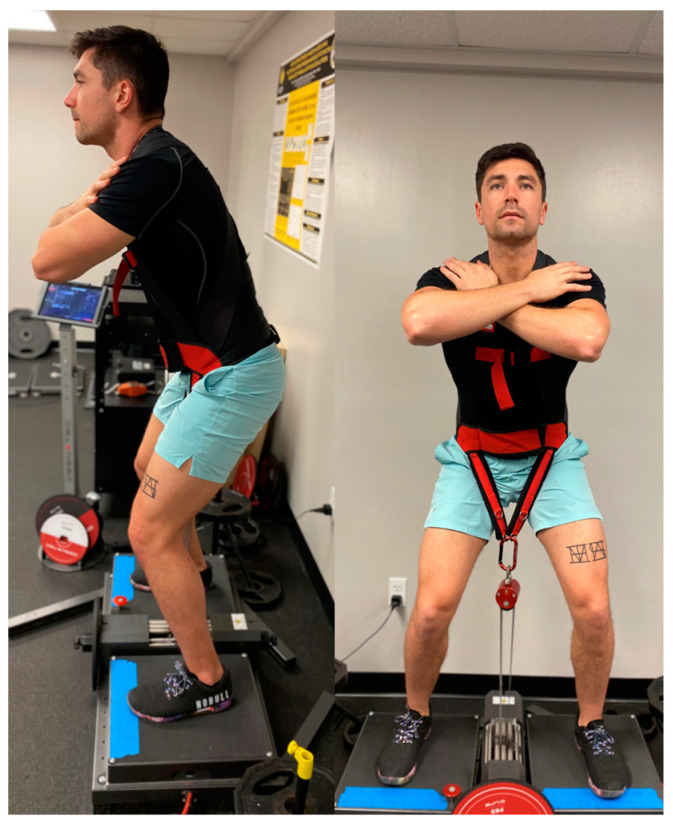
Desmotec Full-Body Harness.

**Figure 2 jfmk-09-00165-f002:**
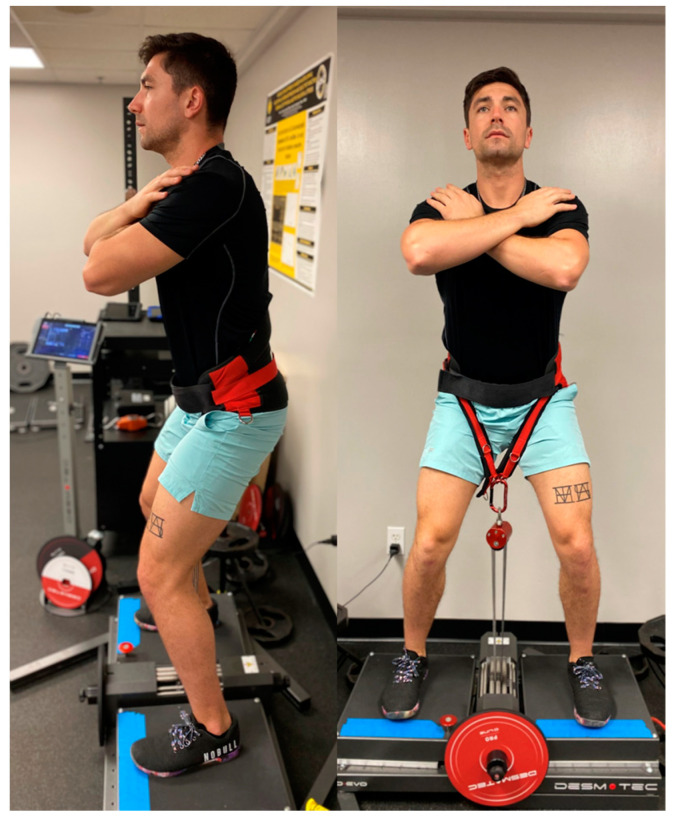
Desmotec Waist Belt.

**Figure 3 jfmk-09-00165-f003:**
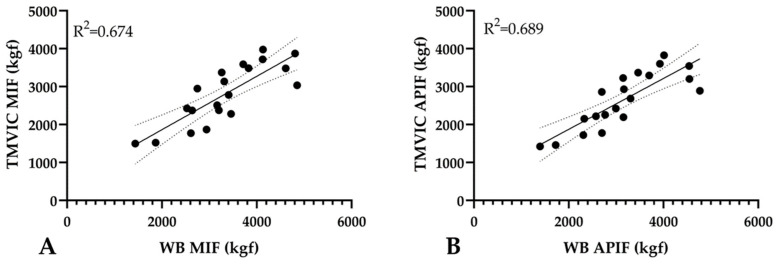
(**A**,**B**) Relationships between WB and TMVIC force variables. TMVIC = traditional maximum isometric contraction, WB = waist belt, MIF = maximum isometric force, and APIF = average peak isometric force. A solid line represents the best fit from linear regression, while dashed lines represent 95% CI. All statistical procedures were accepted as significant at an alpha level of *p* ≤ 0.05.

**Figure 4 jfmk-09-00165-f004:**
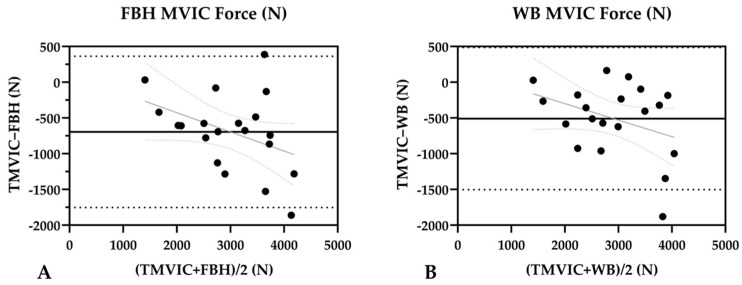
(**A**,**B**) Analysis of agreement between TMVIC and FBH APIF and TMVIC and WB APIF. TMVIC = traditional maximum voluntary isometric contraction; FBH = full-body harness; WB = waist belt; APIF = average peak isometric force. The middle solid black line represents the mean of the difference between the methods for FBH = −695 and for WB = −509 (bias). The upper and lower black dashed lines represent the bias ± 1.96 standard deviations (95% limits of agreement). The straight gray line represents the line of regression, and the gray curved dashed lines represent the 95% confidence intervals of the regression.

**Table 1 jfmk-09-00165-t001:** Isometric force output and pain for the two testing sessions.

Variable	Session 1 (V4)	Session 2 (V5)
FBH MIF (N)	3582 ± 985	3554 ± 1016
WB MIF (N)	3324 ± 914	3332 ± 910
FBH APIF (N)	3321 ± 960	3347 ± 934
WB APIF (N)	3142 ± 861	3161 ± 908
FBH MP	33.85 ± 24.16	35.20 ± 24.08
WB MP	39.10 ± 25.16	38.85 ± 22.58
FBH AP	28.82 ± 21.05	30.22 ± 20.51
WB AP	33.72 ± 22.86	34.20 ± 20.19

Data are presented as mean ± standard deviation. Pain is reported on a scale of 0–100, with 0 indicating “no pain at all” and 100 indicating “maximal or excruciating”. FBH = full-body harness, WB = waist belt, MIF = maximum isometric force, APIF = average peak isometric force of three contractions within a single testing session, MP = maximum subjective pain, and AP = average subjective pain from the three contractions within a single testing session. There are no significant differences between or within testing sessions to report.

**Table 2 jfmk-09-00165-t002:** Reliability data.

Variable	ICC Between Sessions	*p*-Value	ICC (95% CI)	ICC Strength	SEM	MD
FBH MIF	1 and 2 (V4 and V5)	0.609	0.954 (0.889–0.982)	Good to Excellent	22.225	61.604
WB MIF	1 and 2 (V4 and V5)	0.875	0.967 (0.920–0.993)	Excellent	17.174	47.603
FBH APIF	1 and 2 (V4 and V5)	0.694	0.977 (0.940–0.990	Excellent	21.407	59.337
WB APIF	1 and 2 (V4 and V5)	0.683	0.986 (0.964–0.994)	Excellent	15.387	42.649

Two true familiarization sessions were completed prior to session 1. FBH = full-body harness, WB = waist belt, MIF-maximum isometric force, APIF = average peak isometric force of 3 contractions within a single testing session, ICC = Intraclass Correlation Coefficient, SEM = Standard Error of Measurement, and MD = Minimal Difference. All statistical procedures were accepted as significant at an alpha level of *p* ≤ 0.05.

**Table 3 jfmk-09-00165-t003:** Associations between TMVIC outputs with FBH and WB outputs.

Primary Variable	Secondary Variable	r	Strength of Association	R^2^	*p*-Value
TMVIC MIF	FBH MIF	0.813	Very Large	0.661	<0.001
	WB MIF	0.821	Very Large	0.674	<0.001
TMVIC APIF	FBH APIF	0.819	Very Large	0.671	<0.001
	WB APIF	0.830	Very Large	0.689	<0.001
FBH Pain	FBH MIF	−0.060	Trivial	0.004	0.800
	FBH APIF	−0.126	Small	0.016	0.597
FBH AP	FBH MIF	−0.086	Trivial	0.007	0.717
	FBH APIF	−0.145	Small	0.002	0.543
WB Pain	WB MIF	0.024	Trivial	0.000	0.919
	WB APIF	−0.013	Trivial	<0.000	0.958
WB AP	WB MIF	−0.046	Trivial	0.002	0.848
	WB APIF	−0.081	Trivial	0.007	0.733

TMVIC = traditional maximum voluntary isometric contraction, FBH = full-body harness, WB = waist belt, MIF = maximum isometric force, and APIF = average peak isometric force of three contractions within a single testing session. The strength of significant associations (r; *p* ≤ 0.05) between the assessments is as follows: <0.1 = trivial; 0.1 to <0.3 = small; 0.3 to <0.5 = moderate; 0.5 to <0.7 = large; 0.7 to <0.9 = very large; and 0.9 to 1.0 = almost perfect. All statistical procedures were accepted as significant at an alpha level of *p* ≤ 0.05.

**Table 4 jfmk-09-00165-t004:** Results of cross-validation analysis.

Method	Force (N)	CE	r	SEE (N)	SEE%	TE (N)	TE%
TMVIC APIF	2653 ± 734	-	-	-	-	-	-
FBH APIF	3348 ± 939	−694.80	0.819	241.81	7.22	871.69	26.03
WB APIF	3162 ± 909	−508.84	0.830	228.86	7.24	709.72	22.44

Data are presented as mean ± standard deviation. TMVIC = traditional maximum voluntary isometric contraction; FBH = full-body harness; WB = waist belt; APIF = average peak isometric force. SEE% is calculated as SEE/actual_force_, and TE% is calculated as TE/actual_force_.

## Data Availability

The data presented in this study are available upon request from the corresponding author.

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
