# Peer review of "Full-Body Harness versus Waist Belt: An Examination of Force Production and Pain during an Isoinertial Device Maximal Voluntary Isometric Contraction"

_jfmk, 2024, doi:10.3390/jfmk9030165_

Round 1

Reviewer 1 Report (New Reviewer)

Comments and Suggestions for Authors

Thank you for your research and the paper. The study is interesting; however, there are a number of conceptual issues that make that the paper does not reach the scientific interest or sufficient quality to be published.

The objetive of the paper is to compare three different ways of generate de MCV in squat with the resistance on the shoulders (traditional) and with the resistance on the hips, holding the body or waist. You are not usign an inertial device, only the force cells that use this device to measure the force. This device is not validated to measure the force so there is no sense to compare the force measured with it without analyzing previously its validity and reliability with the same position. In this case, positioning the inertial machine under a rack with a fixed bar and measuring for the same person the exercise with the force plate and with the inertial force sensor.

Another posibility could be to compare with an anchor on the ground, the force in an isometric squat with inertial and with the forceplate devices. It would be also interesting to compare for a force plate both positions of squat in force production with both arnesses.

However, in the study  two different movements and two different sensors are mixed, making the conclusions complex. Additionally, pain also appears, generating more confusion in the study.

Going into more detail in the article, the included acronyms make it difficult to follow. More simple acronyms should be used. 

The objetive is not very clear...is the pain or is the force produced. I consider that only one of the should be included as the variable of the study and focus all the study in one of the variables.

The method is not well organized and  very difficult to read. In my opinion the two familiarization sessions should not be considered in the method with an specific number V1 and V2. It only gives confussion to the reader.

The results are not clearly presented. The tables should include the acronyms meaning. Table 3 has errors on the p and R2 values. The legends of the tables are not very clear. 

Author Response

Thank you for your comments on our paper and how to improve it. Please find below a response to all of you comments.

Thank you for your research and the paper. The study is interesting; however, there are a number of conceptual issues that make that the paper does not reach the scientific interest or sufficient quality to be published.

Comment 1: The objetive of the paper is to compare three different ways of generate de MCV in squat with the resistance on the shoulders (traditional) and with the resistance on the hips, holding the body or waist. You are not usign an inertial device, only the force cells that use this device to measure the force. This device is not validated to measure the force so there is no sense to compare the force measured with it without analyzing previously its validity and reliability with the same position. In this case, positioning the inertial machine under a rack with a fixed bar and measuring for the same person the exercise with the force plate and with the inertial force sensor.

Another posibility could be to compare with an anchor on the ground, the force in an isometric squat with inertial and with the forceplate devices. It would be also interesting to compare for a force plate both positions of squat in force production with both arnesses.

Response 1: While your comment is well received, we would like to assert that the rationale of this investigation was to provide a direct examination of the utility of the D.EVO for assessing squat MVIC force if the user did not have access to other equipment. While it is true that validation of the force plates under an immovable bar, or compared to a ground anchor would provide context as to the validity of the force plates themselves, this was not the aim of our investigation. In fact, we argue that this would provide little context for future practical application of the D.EVO. Moreover, if the recommended study was completed, the next logical step would be to complete the investigation presented currently in order to provide usable context to prospective future users. In this investigation we specifically sought to specifically examine if the D.EVO device could be used in lieu of a normal force plate and immovable bar setup (outlined on L402-412). As such, we feel that our direct comparison between MVIC outcomes using either strap (FBH & WB) to a more standard assessment (TMVIC) is appropriate and would be notably more interesting to budget and space conscious laboratories and training rooms as it provides a direct examination of the capabilities of the device for assessing squat MVIC.

Comment 2: However, in the study  two different movements and two different sensors are mixed, making the conclusions complex. Additionally, pain also appears, generating more confusion in the study.

Response 2: We are not sure if you as the reviewer are made aware, but this submission is a resubmission to JFMK. Between submissions we corrected our statistical assessment of validity (as requested) and included additional pain related outcomes (as requested). As such, we are confused as to why the inclusion of pain is now considered to be confusing rather than an informative assessment that provides transparency. While the comparisons may be complex, we would like to reassert that this investigation is simply an examination of whether or not the D.EVO device should be used as a replacement for a traditional force plate set up. As referenced on L43-45, during previous work within our laboratory participants reported feeling pain imposed by the WB. Therefore, considering most researchers and trainers would likely morally and ethically prefer to elicit as little pain as possible (if not the purpose of the investigation) , we felt that comparing force outputs when utilized both straps and presenting the differences in pain imposed by the straps not only promotes transparency but also provide further context as to which strap may be more safely utilized in future investigations.

Comment 3: Going into more detail in the article, the included acronyms make it difficult to follow. More simple acronyms should be used.

Response 3: The acronyms are defined on first mention. However, we would be happy to amend any acronyms that you feel should be removed or made simpler. If the reviewer would like to give more detail regarding specifically which acronyms they feel are confusing, we would be happy to comply. Acronym keys have also been added to each table and figure for clarity.

Comment 4: The objetive is not very clear...is the pain or is the force produced. I consider that only one of the should be included as the variable of the study and focus all the study in one of the variables.

Response 4: As mentioned above, upon a previous review cycle in JFMK we were advised to include more outcomes related to pain. Moreover, we strongly assert that the inclusion of pain outcomes provide greater context and transparency for prospective future users of the device. However, if the reviewer is adamant that these measures must be removed, we will comply.

Comment 5: The method is not well organized and  very difficult to read. In my opinion the two familiarization sessions should not be considered in the method with an specific number V1 and V2. It only gives confussion to the reader.

Response 5: A clearer description of the protocol has been added to L105-107 and reads “This study utilized a within-group repeated measures design with randomization of strap order to evaluate the effects of different IMVIC protocols (FBH and WB) repeated during two different testing sessions (Visit 4 & Visit 5; [V4 & V5]).” To add clarity here, the group assignments just denoted the order in which the FBH or WB IMVICs were completed, with TMVIC always being last. Regardless of group, everyone completed a FBH, WB, and TMVIC. Additionally, added detail has been added to each section of the results to further clarify what visits the outcomes were from.

Comment 6: The results are not clearly presented. The tables should include the acronyms meaning. Table 3 has errors on the p and R2 values. The legends of the tables are not very clear.

Response 6: We are not entirely sure as to which R2 and p values you are referring to in Table 3. We believe the values presented are correct. However, we understand that the >0.001 for the R values could be confusing so we have amended to 0.000 for clarity.

We were previously advised by JFMK reviewers to remove the redundant acronym meanings from the table footnotes. As such, we are confused as to whether or not they should appropriately be included. They have been added back in at the request of this reviewer.

Reviewer 2 Report (New Reviewer)

Comments and Suggestions for Authors

Dear Author, the paper is methodologically well prepared. The introduction, selection of methods, groups of participants and the discussion are correct, but please make necessary changes.

The abstract contains a conclusion that is not supported by the results - I addressed this in the last note below.

Line 228 -  add information about the statistical significance value between FBH MIF and WB MIF, use a posthoc test.

Line 239  - the lack of statistical significance does not mean that the results are the same. It is worth emphasizing the absolute differences between treatment

 Line 245   unnecessary information - “All statistical procedures were accepted as significant 245 at an alpha level of p≤0.05” or add appropriate information in table

 Line 271 - you present results for Relationships between WB and TMVIC force variables why not for FBH?

 Line 274 3.5. Validity – If there are no specific conclusions, it may be worth performing multivariate regression analysis. Maybe this will allow for better prediction of results.

 Line 380 - It is unclear where the conclusion came from. Therefore, we recommend future investigations planning to utilize the D.EVO for assessing IMVIC do so with the WB standard. No such results at work. Justify this conclusion.

Author Response

Thank you for your feedback on our article and your recommendations for its improvement. Please find below a response to your comments.

Comment 1: The abstract contains a conclusion that is not supported by the results - I addressed this in the last note below.

Response 1: The conclusion has been changed and now reads “Our findings indicate that the D.EVO should not be utilized as a replacement for a traditional MVIC setup.”

Comment 2: Line 228 -  add information about the statistical significance value between FBH MIF and WB MIF, use a posthoc test.

Response 2: The posthoc was originally completed and presented an identical p-value p=0.49. However, we have added a hedges g to emphasize the comparison.

Comment 3: Line 239  - the lack of statistical significance does not mean that the results are the same. It is worth emphasizing the absolute differences between treatment

Response 3: Absolute differences are now presented on L247-248 and L251-252.

Comment 4:  Line 245   unnecessary information - “All statistical procedures were accepted as significant 245 at an alpha level of p≤0.05” or add appropriate information in table

Response 4: The described line has been removed.

Comment 5: Line 271 - you present results for Relationships between WB and TMVIC force variables why not for FBH?

Response 5: The linear regression between FBH and TMVIC was not significant. As such we chose not to included the graph as we felt it did not provide any further information and would unnecessarily take up space. Language depicting that only the WB significantly predicted TMVIC via linear regression is included on L76-277 “WB MIF was the only significant predictor of TMVIC MIF (R2=0.674, SEE=46.694) and WB APIF was the only significant predictor of TMVIC APIF (R2=0.688, SEE=42.938).” Further description has been added to L278-279 for added clarity.

Comment 6:  Line 274 3.5. Validity – If there are no specific conclusions, it may be worth performing multivariate regression analysis. Maybe this will allow for better prediction of results.

Response 6: Apologies for out confusion here, but what are you meaning by “no specific conclusions”? We aren’t assessing if the D.EVO can predict TMVIC force perse.  Instead we sought to compare the IMVICs from the D.EVO to TMVIC to see if WB and FBH can be used in conjunction with the force plates and a fixed isoinertial pulley to assess MVIC accurately. We are saying it cannot, but it can be used to reliably assess changes in MVIC across time. Would the reviewer prefer if we used the team “estimate” to make this clearer?

Comment 7: Line 380 - It is unclear where the conclusion came from. Therefore, we recommend future investigations planning to utilize the D.EVO for assessing IMVIC do so with the WB standard. No such results at work. Justify this conclusion.

Response 7: The statement has been removed and replaced with a more appropriate conclusion on L402-407 and reads “Collectively, our findings indicate that the D.EVO device is capable of assessing peak isometric force during a squat at a knee angle of 140° reliably between subsequent testing sessions. However, considering FBH and WB outcomes were not valid predictors of TMVIC outcomes, future outcomes recorded on the D.EVO should only be utilized to assess changes over time rather than being interpreted as a measure of participants true squat MVIC force.”

Round 2

Reviewer 1 Report (New Reviewer)

Comments and Suggestions for Authors

Thank you for your answers. I know that when you submit a paper to a journal you can find different suggestions from the reviewer that sometimes they could be ambiguous. However, as a reviewer, I can not see the previous revisions of the paper so my review does not consider the history of the paper. So, first of all, sorry that the reviewers do not hava a common point of view on different aspects of the article.

I agree with the practial application of the study, but I think that it would have be a second phase and not begining the study with the final application without having previously verified whether the measurments made by this device are valid. 

In my opinion the variable pain only includes confusion in the study and should not be an aim in the study.

The methods is still confussion including 5 visits. It should be clearer to consider only the session 1 and session 2 and not including the rest of the visits listed. You could explain the familiarization protocol and the assesment protocol in a different paragraph. It is not easy to follow. The most important protocol is that of the measurements (V5 and V6) and it should be described without references to other visits ( you must go back in the text to see the information).

Why did you put the traditional squat assesment the third?It could suppose a bias in the measurement by the fatigue of the participants. 

In the results I do not see clear to mix in the same table the pain with the force values.  I think that it is not suitable to apply a two ways ANOVA test. On one side, you can check if there are differences among the three measurements and on the other hand you want to analyze the realibility of the results for the isoinertial device. Mixing in one test both the reliability and the type of system has not sense.  The same is apply to pain. A t test comparing the two type of harness for one of the days or including both days in the analysis could be a better solution. 

The table legends are too long.

Several associations are not easy to understand. Why did not check the pain in traditional method? Maybe the bar could cause pain.

The regression you have applied is not related to the aims of the study and does not add value to the study.

Author Response

Thank you for your answers. I know that when you submit a paper to a journal you can find different suggestions from the reviewer that sometimes they could be ambiguous. However, as a reviewer, I can not see the previous revisions of the paper so my review does not consider the history of the paper. So, first of all, sorry that the reviewers do not hava a common point of view on different aspects of the article.

Comment 1: I agree with the practial application of the study, but I think that it would have be a second phase and not begining the study with the final application without having previously verified whether the measurments made by this device are valid. 

Response 1: We understand your point of view and would like to clarify our rationale. This study is essentially the second phase series of studies aimed at examining the utility of the D.EVO for assessing MVIC and as a stimulus for eliciting muscle damage and measuring the subsequent recovery. The first project of this sequence (submitted for publication elsewhere) indicated that the force plates exhibited excellent test-retest reliability when assessing MVIC one week apart. However, considering the participants  reported pain, which may have influenced their motivation to provide a truly maximal contraction, we felt the next logical step was to examine the utility of another strap (FBH) that would theoretically elicit less pain and thus theoretically make it a better estimate of “true” MVIC force. This is the entire rationale of the current investigation. As such, we assert that this study follows a logical profession and is the next logical step following the findings of our previous investigation. Unfortunately, now that our findings indicate that the FBH and WB are not valid indicators of squat MVIC it is easy to be retroactively skeptical and speculate that another preliminary investigation should have been complete prior. However, we feel that this stance is biased towards our null results since if the current investigation had shown the FBH and WB to be valid, the previous study proposed by the reviewer would not be necessary.

Comment 2: In my opinion the variable pain only includes confusion in the study and should not be an aim in the study.

Response 2: While we understand that it does bring in another element, which may be confusing,  we assert that presenting that the pain is not different between the straps provides valuable context for future users and provides evidence that the pain elicited by the strap is not a confounding variable. Moreover, to highlight your first statement above, both reviewers in out last round of review requested more information and data about the pain be provided for additional context, which we followed.

Comment 3: The methods is still confussion including 5 visits. It should be clearer to consider only the session 1 and session 2 and not including the rest of the visits listed. You could explain the familiarization protocol and the assesment protocol in a different paragraph. It is not easy to follow. The most important protocol is that of the measurements (V5 and V6) and it should be described without references to other visits ( you must go back in the text to see the information).

Response 3: We have separated the screening, familiarization, and testing procedures into different subsections within the experimental design for clarity. We have also added verbiage on lines 104-106 for better clarity regarding the visits and their numbers that reads

“Participants reported to the Exercise Physiology Intervention and Collaboration (EP-IC) lab for a total of 5 visits including 1 screening visit (visit 1; V1) 2 familiarization visits (Visits 2 & 3; [V2 &V3]) and 2 testing sessions (Visit 4 & Visit 5; [V4 & V5]).”

Comment 4: Why did you put the traditional squat assesment the third?It could suppose a bias in the measurement by the fatigue of the participants. 

Response 4: While it is possible that a small amount of fatigue may have occurred, we allowed for 5 minutes rest between MVIC protocols which has been shown to be and adequate rest period to elicit recovery. Additionally, again highlighting your first comment, both reviewers in out previous round were concerned with PAP effects and ensured that the fatigue effects we negligible in comparison. As such we added the previous statement below.

We have addressed this point in the final statement of the limitations “TMVIC always occurred after both FBH and WB IMVIC. Consequentially, small post-activation potentiation (PAP) effects are possible within the current examination. However, we do not believe they are of concern within the present study aim. FBH and WB order was randomized and counterbalanced. Therefore, even if PAP effects were present and augmented TMVIC force, this would not affect the TMVIC’s comparison to the FBH and WS differently.”

We have now amended the statement to read.

“TMVIC always occurred after both FBH and WB IMVIC. Consequentially, small post-activation potentiation (PAP) or fatigue effects are possible within the current examination. However, we do not believe they are of concern within the present study aim. FBH and WB order was randomized and counterbalanced. Therefore, even if PAP or fatigue effects were present and augmented TMVIC force, this would not affect the TMVIC’s comparison to the FBH and WS differently.”

Comment 5: In the results I do not see clear to mix in the same table the pain with the force values.  I think that it is not suitable to apply a two ways ANOVA test. On one side, you can check if there are differences among the three measurements and on the other hand you want to analyze the realibility of the results for the isoinertial device. Mixing in one test both the reliability and the type of system has not sense.  The same is apply to pain. A t test comparing the two type of harness for one of the days or including both days in the analysis could be a better solution. 

Response 5: Apologies for the confusion here but we are having a difficulty understanding your recommendation here, are you suggesting that two-way ANOVAs not be completed whatsoever and replaced with t-tests?

Comment 6: The table legends are too long.

Response 6: The last round of review you stated “The results are not clearly presented. The tables should include the acronyms meaning. “ which we complied with. So we are confused as to what you would prefer as these statements are in direct contradiction to one another. Moreover, it is to our understanding that the tables and figures should stand alone. As such, the relatively long table legends are required to provide appropriate context for the tables.

Comment 7: Several associations are not easy to understand. Why did not check the pain in traditional method? Maybe the bar could cause pain.

Response 7: While indeed we could have checked for pain the in TMVIC, this is the common practice and accepted procedure. Furthermore, in our previous investigation (submitted elsewhere) no participants reported feeling pain during the TMVIC. Therefore, it was out of the scope of this study to re-examine the validity of TMVIC measurement and assess the hypothesized pain imposed by the TMVIC. Also, we are confused as to your statement here as previously you have indicated that we should not include pain.

Comment 8: The regression you have applied is not related to the aims of the study and does not add value to the study.

Response 8: While we understand how this could be seen as redundant. We assert that removing an independent analysis (linear regression) because another independent analysis (total error method) shows null results would be inappropriate and incorrect practice. It is only now that we have presented that the WB is not a valid indicator of TMVIC (via the total error method) that the linear regression be could seen as redundant. As such, we feel that considering it was part of our initial statistical approach, best practice dictates that it be included in the presentation of our findings. Furthermore, we assert that the linear regression continues to provide additional context for this relationship as it characterizes the nature of the relationship between these two variables more clearly than the total error method.

Reviewer 2 Report (New Reviewer)

Comments and Suggestions for Authors

The corrections included are sufficient and better justify the conclusions.

Author Response

Thank you for your guidance in improving our manuscript. 

This manuscript is a resubmission of an earlier submission. The following is a list of the peer review reports and author responses from that submission.

Round 1

Reviewer 1 Report

Comments and Suggestions for Authors

I must appreciate for great manuscript. As a reader/reviewer, I don't have any serious concerns about the manuscript. However, I need to suggest very very few suggestions that will improve the manuscript's future for the novice reader.  

1. The title should be revised for better readability 

2. Subsection 2.3 should be demonstrated with figures. For example, Anthropometrics - What procedure has been followed to measure the body composition? Why figures of Experimental MVIC were not included? MVIC should demonstrate with figures.

3. What is the need of this study?

4. No future? 

Author Response

To the reviewer,

Thank you very much for the kind words about our work. Please find below responses to each of your suggestions.

Comment 1: The title should be revised for better readability. 

Response 1: Title has been revised.

Comment 2: Subsection 2.3 should be demonstrated with figures. For example, Anthropometrics - What procedure has been followed to measure the body composition? Why figures of Experimental MVIC were not included? MVIC should demonstrate with figures.

Response 2: We have added a description of the BIA procedures as well as figures for the MVICs.

Comment 3:  What is the need of this study?

Response 3: We believe we address the need for the study on page 2 lines 80-91 “Several studies to date have investigated the effects of pain, body position, knee angle, and bar position on muscle activation and force production [1,12,17–22]. However, to our knowledge, no studies exist examining the differences in force output during FBH and WB IMVIC’s on the D.EVO or the reliability and validity of a squat FBH IMVIC itself. Moreover, considering pain’s modulatory role in force production, and the reports of pain experienced during WB IMVICs within our own laboratory, an investigation examining the differences in pain, force output, test-retest reliability, and concurrent validity of an FBH IMVIC when compared to a WB IMVIC and TMVIC is warranted.” However, if the reviewer requires further elaboration, we would be willing.

Comment 4: No future? 

Response 4: We believe we address the future implication of our findings on page 9 lines 341-347 “Therefore, we recommend future investigations planning to utilize the D.EVO for assessing IMVIC do so with the standard WB. Collectively, our findings further support the use of the D.EVO for assessing peak isometric force during a squat at a knee angle of 140°. However, in consideration of the pain imposed by the straps during an IMVIC assessment and the device’s relatively low sampling rate (25 Hz), we do not recommend this device as a replacement for more traditional force plate technologies at this time.”. However, if the reviewer However, if the reviewer requires further elaboration, we would be willing.

Reviewer 2 Report

Comments and Suggestions for Authors

General comments

Although there is some interesting data here which would have real implications for practise, there are key methodological flaws, inaccuracies that needs amending. Your understanding of biomechanics and statistics needs work, with incorrect SI units relating to force and wrong statistical approach to determine validity. Additionally, you focus too much upon unpublished work rather than trying to identify any published work within the area to critique too, it is worth noting I reviewed the unpublished work and the same errors were apparent.

Additionally, do you need to focus so much on using the Desmotec devo device, could you not run a validity study to traditional as one study itself and the build upon that with pain, attachment selection and the need for familiarization after?

Specific comment

L30 – MVICs can still come with risk if not administered appropriately.

L30-31 – Based on your methods, I wouldn’t say this was simple to administer.

L36 – provide a reference for the use of isoinertial devices.

Martínez-Hernández, David MSc1,2Flywheel Eccentric Training: How to Effectively Generate Eccentric Overload. Strength and Conditioning Journal 46(2):p 234-250, April 2024. | DOI: 10.1519/SSC.0000000000000795

L37-40 - Additionally, although correct with the terminology over eccentric overload, as it is greater than traditional resistance training it is not supra-maximal load. Be clearer here.

L45 – I would avoid discussing unpublished work within the introduction, a brief mention maybe ok but as with my general comments there is too much of it.

L68 – Provide reference for Park and Hopkins

L75 – remove “our”

L80-81 – What task?

Methods

L101 – As you performed the TMVIC after, do you think there was any PAP effect?

L104 – define BIA on its first use.

L142-143 – Need further information on the DEVO device force assessment, some description of using D-Load cells and configuration would be essential as it currently the reader doesn’t know.

L177 – Any sample size justification based on statistical power?

L185 – What about <0.01?

L187 – What type of ICC? Why use the point estimate and not the lower bound 95% confidence interval which is more robust.

Koo TK, Li MY. A Guideline of Selecting and Reporting Intraclass Correlation Coefficients for Reliability Research. J Chiropr Med. 2016 Jun;15(2):155-63. doi: 10.1016/j.jcm.2016.02.012. Epub 2016 Mar 31. Erratum in: J Chiropr Med. 2017 Dec;16(4):346. PMID: 27330520; PMCID: PMC4913118.

L192-193 – What were the boundaries for correlations based on? Reference required.

Results

-              Force is not measured in Kg, this needs to be clear throughout.

Table 1 – Remove some of the key based text that is repeated and unnecessary.

Table 2 – Why the point estimate? The ICC 2,1, is the wrong model for this study unless you have used multiple testers, if so this needs to be clear within the text.

L239 – You have not assessed validity, you have assessed a relationship between two measures. These are not the same thing.

Figure 1 – Needs improvements, such removing outline, title headings removed, probably include as single figures, ass 95% CIs within the figure would also be useful. Could you develop correction equations based on this.

L257 – You have not assessed validity.

L226-283 – This paragraph is way too much on unpublished work and should not be the key focus of the discussion. You should look for more relevant literature on isometric squat variation of.

L285 – The D.EVO is not the only product you can use an iso-belt squat with, this can be performed in multiple ways, don’t Pidgeon hole yourself with this.

L289 – Reference needed for Hug et al.,

L302-304 – Why did you not present the pain based relationships, it would provide a better story.

L305 – If there were deviations in position, why was this not controlled?

L313 – What about axial loading? But they have identical relationships?

Conclusions

L327 – Not validity.

L328 – Remove all content related to the unpublished work.

L333 – Why use the waist belt if you don’t get a true maximum lower body force production? Think about the implications for performance, prescription and training, such as the dynamic strength index.

Comments on the Quality of English Language

Small edits required in specific comments.

Author Response

To the reviewer,

Thank you for your comments and recommended revisions. Please find below our responses.

Comment 1: Although there is some interesting data here which would have real implications for practise, there are key methodological flaws, inaccuracies that needs amending. Your understanding of biomechanics and statistics needs work, with incorrect SI units relating to force and wrong statistical approach to determine validity. Additionally, you focus too much upon unpublished work rather than trying to identify any published work within the area to critique too, it is worth noting I reviewed the unpublished work and the same errors were apparent.

Response 1: While we agree that we have an overabundance of unpublished work referenced within the article, we are in a rather difficult position as the current article contains an examination that was subsequent to referenced unpublished work. The referenced unpublished precursor work was submitted elsewhere over 10 months ago and was only recently given a decision. Since then, we have addressed its methodological flaws and resubmitted the precursor manuscript elsewhere and are awaiting decision. Once published, it will bring more appropriate context to the current article. However, your point is well received, and the specific unpublished data has been removed from the current article. If further change is reaffirmed we would be willing to amend further. Additionally, we would like to reaffirm that we are perfectly justified in labeling this relationship as concurrent validity as concurrent validity is just examining the relationship of a new test with a previously established test. This can be done so with the use of Pearson’s and a regression. Both of which were used in the current investigation. [2]

Comment 2: Additionally, do you need to focus so much on using the Desmotec devo device, could you not run a validity study to traditional as one study itself and the build upon that with pain, attachment selection and the need for familiarization after?

Response 2: You are essentially describing our current laboratory’s objectives. We are currently undergoing a series of sequential investigations that will utilize the D.EVO device for a number of different research examinations. The referenced precursor unpublished data addresses familiarization and concurrent validity of the D.EVO for isoinertial and isometric squat force. The current investigation was designed as a subsequent analysis following the reports of hip pain imposed by the standard waist belt and is meant to ensure proper methodology moving forward.

Comment 3: L30 – MVICs can still come with risk if not administered appropriately

Response 3: Language added for clarity.

Comment 4: L30-31 – Based on your methods, I wouldn’t say this was simple to administer.

Response 4: While the description within the methods outlines and seemingly rather difficult process, the administration of the D.EVO MVIC really only entails one additional step when compared to a barbell setup. However, your point is well received and  “Administer” has been removed.

Comment 5: L36 – provide a reference for the use of isoinertial devices. Martínez-Hernández, David MSc1,2. Flywheel Eccentric Training: How to Effectively Generate Eccentric Overload. Strength and Conditioning Journal 46(2):p 234-250, April 2024. | DOI: 10.1519/SSC.0000000000000795

Response 5. Added.

Comment 6: L37-40 - Additionally, although correct with the terminology over eccentric overload, as it is greater than traditional resistance training it is not supra-maximal load. Be clearer here.

Response 6: Language added for clarity.

Comment 7: L45 – I would avoid discussing unpublished work within the introduction, a brief mention maybe ok but as with my general comments there is too much of it.

Response 7: Same response as response 1 above.

Comment 8: L68 – Provide reference for Park and Hopkins

Response 8: Added.

Comment 9: L75 – remove “our”

Response 9: Removed

Comment 10: L80-81 – What task?

Response 10: Language has been added for clarity.

Comment 11: L101 – As you performed the TMVIC after, do you think there was any PAP effect?

Response 11: While small PAP effects are possible in this scenario, we do not believe they are of concern within the present study aim. Considering FBH and WB order was randomized and counter-balanced, even if PAP effects were present and augmented TMVIC force, this would not affect the TMVIC’s comparison to the FBH and WS differently. Additionally, we are not making any comparisons of the TMVIC to other like TMVICs from different investigations for the purpose of establishing concurrent validity. Therefore, we do not consider the possible presence of PAP effects to be of concern for the current study’s aim and findings.

Comment 12: L104 – define BIA on its first use.

Response 12: Defined.

Comment 13: L142-143 – Need further information on the DEVO device force assessment, some description of using D-Load cells and configuration would be essential as it currently the reader doesn’t know.

Response 13: Language added for clarity.

Comment 14: L177 – Any sample size justification based on statistical power?

Response 14: Yes, a power analysis using power analysis software (G*Power 3.1.9.4, HHU, Dusseldorf, Germany) revealed that to detect a moderate Pearson’s correlation (r = .60) with a power of .80 and p-value of 0.05, a sample size of 19 would be required. Language has been added to page 4 lines 123-125.

Comment 15: L185 – What about <0.01?

Response 15: Changed.

Comment 16: L187 – What type of ICC? Why use the point estimate and not the lower bound 95% confidence interval which is more robust.

Response 16: We have changed our presentation of ICCs within table 2. Additionally, while the lower bound of the 95% CI provides the most robust interpretation, we contend that this practice is overly conservative. Further, we assert that when interpreting ICCs, the point estimate provides the most practical interpretation of the reliability as this is the most probable value. When based on the lower bound, you are instead essentially stating that you are 95% certain the value is above that mark rather than giving the most probable mark. However, if the reviewer further asserts that our practice is unacceptable, we are willing to amend.

Comment 17: L192-193 – What were the boundaries for correlations based on? Reference required.

Response 17: Boundaries changed, and Citation added.

Comment 18: Force is not measured in Kg, this needs to be clear throughout.

Response 18: Changed.

Comment 19: Table 1 – Remove some of the key based text that is repeated and unnecessary.

Response 19: Removed.

Comment 20: Table 2 – Why the point estimate? The ICC 2,1, is the wrong model for this study unless you have used multiple testers, if so, this needs to be clear within the text.

Response 20: In accordance with the Koo article provided [1] we utilized the Two-way mixed effects, absolute agreement multiple measurement ICC method. We have changed the table description for clarity.

Comment 21: L239 – You have not assessed validity, you have assessed a relationship between two measures. These are not the same thing.

Response 21: Language has been changed to a more correct verbiage “concurrent validity”. We would like to reaffirm that we are perfectly justified in labeling this relationship as concurrent validity as concurrent validity is just examining the relationship of a new test with a previously established test. This can be done so sufficiently with the use of Pearson’s and/or a regression. Both of which were used in the current investigation. [2]

Comment 22: Figure 1 – Needs improvements, such removing outline, title headings removed, probably include as single figures, ass 95% CIs within the figure would also be useful. Could you develop correction equations based on this.

Response 22: Figures improved with 95% CI intervals added. Correction equations fall outside the scope of the study and therefore we feel would not be appropriate to include.

Comment 23: L257 – You have not assessed validity.

Response 23: Language amended to concurrent. Also addressed in response 1.

Comment 24: L226-283 – This paragraph is way too much on unpublished work and should not be the key focus of the discussion. You should look for more relevant literature on isometric squat variation of.

Response 24: Addressed in response 1. Additionally, while indeed it would provide more context to the discussion at large, this investigation was conducted specifically as a subsequent investigation and therefore we feel comparisons are most appropriately made to the precursor work. Again, we would preferred to have had that precursor work published by this point to provide a more appropriate reference but as it stand this unpublished work remains the most directly related point of discussion considering the limited amount of work utilizing a Desmotec D.EVO.

Comment 25: L285 – The D.EVO is not the only product you can use an iso-belt squat with, this can be performed in multiple ways, don’t Pidgeon hole yourself with this.

Response 25: We are not meaning to Pidgeon hole ourselves but rather to be specific to the device currently utilized as there is limited data available regarding the agreement of analogous devices when assessing isometric force during a squat.

Comment 26: L289 – Reference needed for Hug et al.,

Response 26: Added.

Comment 27: L302-304 – Why did you not present the pain based relationships, it would provide a better story.

Response 27: Pain-based relationships have been added to Table 3.

Comment 28: L305 – If there were deviations in position, why was this not controlled?

Response 28: The deviations in body position between the two straps is referring to the slight change in posture elicited by the straps themselves. Because of how the straps were situated on the individuals i.e. the over the shoulder straps of the FBH, and where the anchoring carabiner was positioned on the strap itself, the difference in body position was unavoidable and therefore an inherent difference between the two straps. Accordingly, we felt it best to allow participants to select a position of comfort for each strap as opposed to standardizing the hips, shoulders etc., as this would have like changed natural biomechanics. Additionally, the orientation and dimensions of the FBH made it virtually impossible to perfectly mimic the position of the WB.

Comment 29: L313 – What about axial loading? But they have identical relationships?

Response 29: A discussion of the axial loading of the FBH has been added to page 8 lines 306-309.

Comment 30: L327 – Not validity.

Response 30: Addressed in response 1.

Comment 31: L328 – Remove all content related to the unpublished work.

Response 31: Same as Response 1.

Comment 32: L333 – Why use the waist belt if you don’t get a true maximum lower body force production? Think about the implications for performance, prescription and training, such as the dynamic strength index.

Response 32: We believe that you are getting a truly maximal force production. As we address on page 8 lines 312-321 “. One explanation for this phenomenon may be the slight deviations in body position be-tween the FBH and WB during their respective IMVICs. Previously it has been shown that the low-bar back squat position evokes different joint angles than that of a high-bar back squat position which may subsequently increase its load bearing capacity [20]. Within the current investigation, the over-the-shoulder straps of the FBH elicited a forward leaning posture during the IMVIC as opposed to the upright posture of the WB IMVIC. Although slight, this deviation in joint angles may have made the FBH IMVIC more closely representative of a low-bar back squat and explain why FBH MIF were greater than WB MIF. This in turn may have limited the ability of the FBH IMVIC to predict force in the TMVIC assessment, which was more akin to a high-bar squat.” We assert that body position differences between straps account for the force production differences and therefore, at the body position elicited by the WB, we are attaining maximal force.

REFERENCES

  1. Koo, T.K.; Li, M.Y. A Guideline of Selecting and Reporting Intraclass Correlation Coefficients for Reliability Research. Journal of Chiropractic Medicine 2016, 15, 155–163, doi:10.1016/j.jcm.2016.02.012.
  2. Blazevich, A.J.; Gill, N.; Newton, R.U. Reliability and Validity of Two Isometric Squat Tests. J Strength Cond Res 2002, 16, 298, doi:10.1519/1533-4287(2002)016<0298:RAVOTI>2.0.CO;2.

Round 2

Reviewer 2 Report

Comments and Suggestions for Authors

Dear authors,

Thank you for making the extensive changes and revisions, I also appreciate the dialogue as this is important for the learning process (for both parties). Just as a side note, to aid in the review process when resubmitting work I would suggest highlighting changes made within the text, either changing font colour or using the highlight tool. Additionally, providing the line numbers is also useful when amending text.

 Please see responses to each comment below

Comment 1: Although there is some interesting data here which would have real implications for practise, there are key methodological flaws, inaccuracies that needs amending. Your understanding of biomechanics and statistics needs work, with incorrect SI units relating to force and wrong statistical approach to determine validity. Additionally, you focus too much upon unpublished work rather than trying to identify any published work within the area to critique too, it is worth noting I reviewed the unpublished work and the same errors were apparent.

Response 1: While we agree that we have an overabundance of unpublished work referenced within the article, we are in a rather difficult position as the current article contains an examination that was subsequent to referenced unpublished work. The referenced unpublished precursor work was submitted elsewhere over 10 months ago and was only recently given a decision. Since then, we have addressed its methodological flaws and resubmitted the precursor manuscript elsewhere and are awaiting decision. Once published, it will bring more appropriate context to the current article. However, your point is well received, and the specific unpublished data has been removed from the current article. If further change is reaffirmed we would be willing to amend further. Additionally, we would like to reaffirm that we are perfectly justified in labeling this relationship as concurrent validity as concurrent validity is just examining the relationship of a new test with a previously established test. This can be done so with the use of Pearson’s and a regression. Both of which were used in the current investigation. [2]

 Although I appreciate research is a difficult balance and trying “periodize” submissions for sequential flow of publications, the overall process of research is inconsistent. Therefore, it is important not to base future works on previous submissions alone there needs to be a background of relevant literature from other sources or areas. For instance, the current study could have used research looking the difference between other isometric variants (e.g. isometric squat vs iso-pull) which is one which includes axial loading and one which doesn’t (See reference below). I look forward to seeing the reworked article published, but again you can not base your sole rational for the present study on an unpublished study suggesting that when that is published it will provide context.

Brady CJ, Harrison AJ, Flanagan EP, Haff GG, Comyns TM. A Comparison of the Isometric Midthigh Pull and Isometric Squat: Intraday Reliability, Usefulness, and the Magnitude of Difference Between Tests. Int J Sports Physiol Perform. 2018 Aug 1;13(7):844-852. doi: 10.1123/ijspp.2017-0480. Epub 2018 Jul 28. PMID: 29182457.

Although I understand your comment around validity and a regression is certainly suitable statistical style of test for that matter, however, you have only used variables which have presented significant correlations. But a Pearson’s correlation only identifies if two measures are linearly related, it does not provide any indication of systematic errors of bias’s between devices. Therefore, Pearson’s correlations should be avoided and the use of regressions such as Ordinary least squares regressions or Orthogonal regressions. If the authors are adamant to continue to use the Pearson’s correlations, I would suggest present Bland Altman plots to show the visual representation of any Bias between measurements.

Karras DJ. Statistical methodology: II. Reliability and variability assessment in study design, Part A. Acad Emerg Med. 1997 Jan;4(1):64-71. doi: 10.1111/j.1553-2712.1997.tb03646.x. PMID: 9110015.

Kane, M. T., & Mroch, A. A. (2010). Modeling Group Differences in OLS and Orthogonal Regression: Implications for Differential Validity Studies. Applied Measurement in Education, 23(3), 215–241. https://doi.org/10.1080/08957347.2010.485990

Bland , J. M. and Altman , D. G. 1986 . Statistical methods for assessing agreement between two methods of clinical measurement . The Lancet , 1 : 307 – 310 .

Comment 2: Additionally, do you need to focus so much on using the Desmotec devo device, could you not run a validity study to traditional as one study itself and the build upon that with pain, attachment selection and the need for familiarization after?

Response 2: You are essentially describing our current laboratory’s objectives. We are currently undergoing a series of sequential investigations that will utilize the D.EVO device for a number of different research examinations. The referenced precursor unpublished data addresses familiarization and concurrent validity of the D.EVO for isoinertial and isometric squat force. The current investigation was designed as a subsequent analysis following the reports of hip pain imposed by the standard waist belt and is meant to ensure proper methodology moving forward.

 This makes more sense, but I think other companies, products and performance assessments will have similar issues and maybe doesn’t need be as forefront. Or provide stronger conclusions of these implications.

Comment 3: L30 – MVICs can still come with risk if not administered appropriately

Response 3: Language added for clarity.

 Ok

Comment 4: L30-31 – Based on your methods, I wouldn’t say this was simple to administer.

Response 4: While the description within the methods outlines and seemingly rather difficult process, the administration of the D.EVO MVIC really only entails one additional step when compared to a barbell setup. However, your point is well received and  “Administer” has been removed.

 Ok

Comment 5: L36 – provide a reference for the use of isoinertial devices. Martínez-Hernández, David MSc1,2. Flywheel Eccentric Training: How to Effectively Generate Eccentric Overload. Strength and Conditioning Journal 46(2):p 234-250, April 2024. | DOI: 10.1519/SSC.0000000000000795

Response 5. Added.

 Ok

Comment 6: L37-40 - Additionally, although correct with the terminology over eccentric overload, as it is greater than traditional resistance training it is not supra-maximal load. Be clearer here.

Response 6: Language added for clarity.

 Ok 

Comment 7: L45 – I would avoid discussing unpublished work within the introduction, a brief mention maybe ok but as with my general comments there is too much of it.

Response 7: Same response as response 1 above.

Better as you have limited the use of unpublished work.

Comment 8: L68 – Provide reference for Park and Hopkins

Response 8: Added.

  Ok 

Comment 9: L75 – remove “our”

Response 9: Removed

 Ok  

Comment 10: L80-81 – What task?

Response 10: Language has been added for clarity.

 Ok  

Comment 11: L101 – As you performed the TMVIC after, do you think there was any PAP effect?

Response 11: While small PAP effects are possible in this scenario, we do not believe they are of concern within the present study aim. Considering FBH and WB order was randomized and counter-balanced, even if PAP effects were present and augmented TMVIC force, this would not affect the TMVIC’s comparison to the FBH and WS differently. Additionally, we are not making any comparisons of the TMVIC to other like TMVICs from different investigations for the purpose of establishing concurrent validity. Therefore, we do not consider the possible presence of PAP effects to be of concern for the current study’s aim and findings.

 Ok, I would briefly add some of this to the methods with the counterbalancing it would have similar PAP response.

Comment 12: L104 – define BIA on its first use.

Response 12: Defined.

 Ok  

Comment 13: L142-143 – Need further information on the DEVO device force assessment, some description of using D-Load cells and configuration would be essential as it currently the reader doesn’t know.

Response 13: Language added for clarity.

 Ok

Comment 14: L177 – Any sample size justification based on statistical power?

Response 14: Yes, a power analysis using power analysis software (G*Power 3.1.9.4, HHU, Dusseldorf, Germany) revealed that to detect a moderate Pearson’s correlation (r = .60) with a power of .80 and p-value of 0.05, a sample size of 19 would be required. Language has been added to page 4 lines 123-125.

What literature have you used to determine a moderate correlation? This needs referencing appropriately.

Comment 15: L185 – What about <0.01?

Response 15: Changed.

Ok

Comment 16: L187 – What type of ICC? Why use the point estimate and not the lower bound 95% confidence interval which is more robust.

Response 16: We have changed our presentation of ICCs within table 2. Additionally, while the lower bound of the 95% CI provides the most robust interpretation, we contend that this practice is overly conservative. Further, we assert that when interpreting ICCs, the point estimate provides the most practical interpretation of the reliability as this is the most probable value. When based on the lower bound, you are instead essentially stating that you are 95% certain the value is above that mark rather than giving the most probable mark. However, if the reviewer further asserts that our practice is unacceptable, we are willing to amend.

 You have suggested you used the methods by Koo and Li, however, Koo and Li suggest using the lower bound 95% confidence interval therefore you cannot suggest you this if you have not. The assertion by the authors around being overly conservative is inappropriate in this instance, the use of the 95% confidence interval provides an actual meaning to the reliability in what researchers or other practitioners can use. Additionally, as you have not ran a-priori sample size estimations for the present reliability aspects, which will likely be greater than the null hypothesis sample size estimations, you are likely underpowered hence the further need for the use of the 95% CI

Kottner, J., Audige, L., Broson, S., Donner, A., Gajewski, B. J., Hrobjartsson, A., Roberts, C.,Shoukir, M., & Streiner, D. L. (2011). Guidelines for reporting reliability and agreement studies(GRRAS) were proposed. International Journal of Nursing Studies, 48(6), 661–671

McGraw, K. O., & Wong, S. P. (1996). Forming inferences about some intraclass correlationcoefficients. Psychological Methods, 1(1), 30–46.

Shrout, P. E., & Fleiss, J. L. (1979). Intraclass correlations: Uses in assessing rater reliability.Psychological Bulletin, 86(2), 420–428.

Hazra A. Using the confidence interval confidently. J Thorac Dis. 2017 Oct;9(10):4125-4130. doi: 10.21037/jtd.2017.09.14. PMID: 29268424; PMCID: PMC5723800.

Borg DN, Bach AJE, O'Brien JL, Sainani KL. Calculating sample size for reliability studies. PM&R. 2022; 14(8): 1018-1025. doi:10.1002/pmrj.12850

Mokkink, L.B., de Vet, H., Diemeer, S. et al. Sample size recommendations for studies on reliability and measurement error: an online application based on simulation studies. Health Serv Outcomes Res Method 23, 241–265 (2023). https://doi.org/10.1007/s10742-022-00293-9

Comment 17: L192-193 – What were the boundaries for correlations based on? Reference required.

Response 17: Boundaries changed, and Citation added.

OK

Comment 18: Force is not measured in Kg, this needs to be clear throughout.

Response 18: Changed.

Ok, but this this is still inappropriate for force and not the SI unit. It needs converting to Newtons if this is the units provided by the DEVO.

Comment 19: Table 1 – Remove some of the key based text that is repeated and unnecessary.

Response 19: Removed.

OK

Comment 20: Table 2 – Why the point estimate? The ICC 2,1, is the wrong model for this study unless you have used multiple testers, if so, this needs to be clear within the text.

Response 20: In accordance with the Koo article provided [1] we utilized the Two-way mixed effects, absolute agreement multiple measurement ICC method. We have changed the table description for clarity.

OK, please see previous comments on CI.

Comment 21: L239 – You have not assessed validity, you have assessed a relationship between two measures. These are not the same thing.

Response 21: Language has been changed to a more correct verbiage “concurrent validity”. We would like to reaffirm that we are perfectly justified in labeling this relationship as concurrent validity as concurrent validity is just examining the relationship of a new test with a previously established test. This can be done so sufficiently with the use of Pearson’s and/or a regression. Both of which were used in the current investigation. [2]

 Please see previous comment on Pearsons for validity.

Comment 22: Figure 1 – Needs improvements, such removing outline, title headings removed, probably include as single figures, ass 95% CIs within the figure would also be useful. Could you develop correction equations based on this.

Response 22: Figures improved with 95% CI intervals added. Correction equations fall outside the scope of the study and therefore we feel would not be appropriate to include.

Ok, nice figures.

Comment 23: L257 – You have not assessed validity.

Response 23: Language amended to concurrent. Also addressed in response 1.

Ok

Comment 24: L226-283 – This paragraph is way too much on unpublished work and should not be the key focus of the discussion. You should look for more relevant literature on isometric squat variation of.

Response 24: Addressed in response 1. Additionally, while indeed it would provide more context to the discussion at large, this investigation was conducted specifically as a subsequent investigation and therefore we feel comparisons are most appropriately made to the precursor work. Again, we would preferred to have had that precursor work published by this point to provide a more appropriate reference but as it stand this unpublished work remains the most directly related point of discussion considering the limited amount of work utilizing a Desmotec D.EVO.

I appreciate that the unpublished work is a precursor to present study, but if it is that important you should have waited till that work is published. In my opinion it is not appropriate to focus so much on the unpublished work, which has not been accepted via the peer review process. I would suggest only briefly mentioning this as a side not to other related published work, such as Brady et al.

Comment 25: L285 – The D.EVO is not the only product you can use an iso-belt squat with, this can be performed in multiple ways, don’t Pidgeon hole yourself with this.

Response 25: We are not meaning to Pidgeon hole ourselves but rather to be specific to the device currently utilized as there is limited data available regarding the agreement of analogous devices when assessing isometric force during a squat.

I understand this point, but one of the present study, was to look at the pain related response on different harnesses. Having piloted some work around this myself when not using the D.EVO highlighting further implication for other who probably won’t use this device would be crucial.

Comment 26: L289 – Reference needed for Hug et al.,

Response 26: Added.

OK

Comment 27: L302-304 – Why did you not present the pain based relationships, it would provide a better story.

Response 27: Pain-based relationships have been added to Table 3.

OK

Comment 28: L305 – If there were deviations in position, why was this not controlled?

Response 28: The deviations in body position between the two straps is referring to the slight change in posture elicited by the straps themselves. Because of how the straps were situated on the individuals i.e. the over the shoulder straps of the FBH, and where the anchoring carabiner was positioned on the strap itself, the difference in body position was unavoidable and therefore an inherent difference between the two straps. Accordingly, we felt it best to allow participants to select a position of comfort for each strap as opposed to standardizing the hips, shoulders etc., as this would have like changed natural biomechanics. Additionally, the orientation and dimensions of the FBH made it virtually impossible to perfectly mimic the position of the WB.

Add some of this information to the manuscript as it is useful, additionally citing literature looking at differences in trunk angle during the mid-thigh pull could be useful as well to support this.

ʼSantos T, Thomas C, Jones PA, McMahon JJ, Comfort P. The Effect of Hip Joint Angle on Isometric Midthigh Pull Kinetics. J Strength Cond Res. 2017 Oct;31(10):2748-2757. doi: 10.1519/JSC.0000000000002098. PMID: 28933711.

Comfort, P., Jones, P. A., McMahon, J. J., & Newton, R. (2015). Effect of Knee and Trunk Angle on Kinetic Variables During the Isometric Midthigh Pull: Test–Retest Reliability. International Journal of Sports Physiology and Performance, 10(1), 58-63. Retrieved May 24, 2024, from https://doi.org/10.1123/ijspp.2014-0077

Beckham GK, Sato K, Santana HAP, Mizuguchi S, Haff GG, Stone MH. Effect of Body Position on Force Production During the Isometric Midthigh Pull. J Strength Cond Res. 2018 Jan;32(1):48-56. doi: 10.1519/JSC.0000000000001968. PMID: 28486331.

Comment 29: L313 – What about axial loading? But they have identical relationships?

Response 29: A discussion of the axial loading of the FBH has been added to page 8 lines 306-309.

 Ok

Comment 30: L327 – Not validity.

Response 30: Addressed in response 1.

Ok, see previous responses.

Comment 31: L328 – Remove all content related to the unpublished work.

Response 31: Same as Response 1.

Ok, see previous responses.

Comment 32: L333 – Why use the waist belt if you don’t get a true maximum lower body force production? Think about the implications for performance, prescription and training, such as the dynamic strength index.

Response 32: We believe that you are getting a truly maximal force production. As we address on page 8 lines 312-321 “. One explanation for this phenomenon may be the slight deviations in body position be-tween the FBH and WB during their respective IMVICs. Previously it has been shown that the low-bar back squat position evokes different joint angles than that of a high-bar back squat position which may subsequently increase its load bearing capacity [20]. Within the current investigation, the over-the-shoulder straps of the FBH elicited a forward leaning posture during the IMVIC as opposed to the upright posture of the WB IMVIC. Although slight, this deviation in joint angles may have made the FBH IMVIC more closely representative of a low-bar back squat and explain why FBH MIF were greater than WB MIF. This in turn may have limited the ability of the FBH IMVIC to predict force in the TMVIC assessment, which was more akin to a high-bar squat.” We assert that body position differences between straps account for the force production differences and therefore, at the body position elicited by the WB, we are attaining maximal force.

This doesn’t make sense; in your methods you state consistent knee angles of 140 degrees at the knee. You have not cited any of the previous literature on the effect of the trunk angle on kinetics, so I would assume a similar trunk (with slight deviation of a few degrees as highlighted) would be present. Even in your images provided with the test, the change in trunk angle is minimal although you have avoided a full sagittal plane image for the full body harness. Therefore, if the waist belt does not achieve the same force output as the full body harness and it is not due to the changes in knee angle. It could be related to the differences in trunk angle given the literature provided above, but the large deviations reported in those study would not be classed as small deviations. Therefore, it is the difference in attachment type impacting the force production, with the full body harness providing greater force output which will have implications on the application of a waist belt assessment.

ʼSantos T, Thomas C, Jones PA, McMahon JJ, Comfort P. The Effect of Hip Joint Angle on Isometric Midthigh Pull Kinetics. J Strength Cond Res. 2017 Oct;31(10):2748-2757. doi: 10.1519/JSC.0000000000002098. PMID: 28933711.

Comfort, P., Jones, P. A., McMahon, J. J., & Newton, R. (2015). Effect of Knee and Trunk Angle on Kinetic Variables During the Isometric Midthigh Pull: Test–Retest Reliability. International Journal of Sports Physiology and Performance, 10(1), 58-63. Retrieved May 24, 2024, from https://doi.org/10.1123/ijspp.2014-0077

Beckham GK, Sato K, Santana HAP, Mizuguchi S, Haff GG, Stone MH. Effect of Body Position on Force Production During the Isometric Midthigh Pull. J Strength Cond Res. 2018 Jan;32(1):48-56. doi: 10.1519/JSC.0000000000001968. PMID: 28486331.

Comments on the Quality of English Language

Some minor changes needed, but generally a high quality throughout.